# Efficacy of Bottle Gourd Seeds’ Extracts in Chemical Hazard Reduction Secreted as Toxigenic Fungi Metabolites

**DOI:** 10.3390/toxins13110789

**Published:** 2021-11-08

**Authors:** Adel G. Abdel-Razek, Ahmed N. Badr, Salman S. Alharthi, Khaled A. Selim

**Affiliations:** 1Fats and Oils Department, National Research Centre, Dokki, Cairo 12622, Egypt; adelgabr2@gmail.com; 2Food Toxicology and Contaminants Department, National Research Centre, Dokki, Cairo 12622, Egypt; 3Chemistry Department, College of Science, Taif University, P.O. Box 11099, Taif 21944, Saudi Arabia; s.a.alharthi@tu.edu.sa; 4Food Science and Technology Department, Faculty of Agriculture, Fayoum University, Fayoum 6351, Egypt; kas00@fayoum.edu.eg

**Keywords:** antifungal activity, bioactive components, bottle gourd seeds extract, extract toxicity and antitoxic, mycotoxin, toxigenic fungi

## Abstract

Bottle gourd seeds are surrounded by innumerable bioactive components of phytochemicals. This work aimed to evaluate the effectiveness of bottle gourd extracts as antimicrobial and an-ti-mycotoxigenic against toxigenic fungi and mycotoxins. Polar and nonpolar extracts were made from the seeds. The polar eco-friendly extract was prepared by an ultrasonication-assisted technique utilizing aqueous isopropanol (80%), whereas the non-polar extract was obtained using petroleum ether (40–60). The antioxidant efficacy, total phenolic content, and flavonoid content of the extracts were all measured. The fatty acid profile was measured using GC equipment, and the influence on toxigenic fungus and mycotoxin release was also investigated. The antioxidant efficacy of the polar extract is reflected. The total phenolic values of the oil and polar extract were 15.5 and 267 mg of GAE/g, respectively. The total flavonoid content of the oil was 2.95 mg catechol/g, whereas the isopropyl extract of seeds contained 14.86 mg catechol/g. The polar extract inhibited the DPPH more effectively than oil. When compared to other seed oils, the fatty acid composition differed. The pathogens were distinguished by the MIC and MFC for the polar extract. Three sterols were found in the oil, with a high concentration of B-sitosterols. The oil’s valuable -carotene content and tocopherol content were recorded. When compared to traditional antibiotics, the polar extract has shown promising antimicrobial activity against infections and toxigenic fungi. Bottle gourd extracts, as a non-traditional bioactive source, are viewed as a potentially promising alternative that might contribute to increased food safety, shelf-life, and security.

## 1. Introduction

Natural products and phytochemicals are attracting the interest of the universal responsibility for bacterial resistance, and the rising occurrence of oxidative stress scenarios leads to chronic diseases [1,2]. Phytochemicals provide several advantages in resistance to infectious and cancerous problems. These issues could be joined to the production conditions. Plants previously practiced finding novel bioactive constituents. The utilization of herbal plants for the medication is often practiced by up to 90% of the planet’s population, particularly in Africa [3]. Likewise, the pharmaceutical sector developed approximately 61% of novel natural product-based medications that were extremely effective against infectious and cancerous illnesses [4].

Traditional medicines are the best and most cost-effective source of anti-infection and anti-cancer chemicals, according to the world health organization [5]. As a result, one of the main objectives is to screen expected natural compounds and phytochemicals that can be used to decrease infection and cytotoxic effects in food products contaminated with mycotoxins. Physiologically active chemicals in the Cucurbitaceae family include phenolic, steroid, glucosides, and alkaloids [6], conjugated linolenic acid, and its isomers [7], lyso-phosphatidylcholine [8], organo-sulfur fragments, and terpenoids have all been identified as physiologically active compounds. Phytochemicals and compounds derived from the gourd family have been studied to treat certain serious illnesses, including microbial invasion, human immunodeficiency virus, tumor initiators, and cancer [9,10].

Mycotoxins are harmful compounds secreted by fungal contamination of food products [11]. It happens during the pre-harvest or post-harvest stages of the planting process. Cross-contamination can happen during the transportation, storage, processing, and handling of agrofood items, as well as during the food processing phases [12]. Recently, plant extracts practice efficiency to reduce mycotoxin hazard while it presents in tissues [13]. Besides, extracts of plants showed a valuable capacity for toxigenic fungal growth (assessed in synthetic media) [14,15]. The promising impacts of wild plants like stevia, *Opuntia Ficus indica* [16,17], and star anise [18] were recorded. Also, extracted seed oils showed a significant ability to inhibit mycotoxin production by their toxigenic fungi strains, among which were jojoba, jatropha [19], black cumin seed [20], pomegranate oil [21], and hibiscus oils [22]. The common factor between both types of seed extracts (polar and non-polar) is the high quantities of antioxidants and phenolic compounds. Regarding the previous investigations, plant extract contents of bioactive substances, particularly phenolic, have a pivotal function in the reduction of mycotoxin production.

Bottle gourd (BG) is one of the veggies and calorie-dense that dieticians recommend in weight-loss regimens [23]. The spongy flesh tissues along with white pulp and embedded seeds exist inside the bottle gourd fruits. The seeds are present in large numbers, and all are covered with a protectant layer. Seeds are distinguished by their content of protein, fats, dietary fibers, and low carbohydrates [24,25]. Seed kernels mostly yield up to 53% oil as clear and pale yellowish used for cooking and hair oil. The flour of seeds is considered a significant source of vitamins (B-complex vitamins) and minerals inclusive of potassium, calcium, zinc, magnesium, iron, and manganese [25].

Normally, the seeds contain numerous compounds, which ensured food to the germ and provide its protection [26]. Bottle gourd seeds may be distinguished regarding their content of active substances. The present investigation has targeted the determination of the bioactive contents of the bottle-gourd extracts obtained from seeds. Also, to evaluate the potency of these extracts, if applied as antifungal and anti-mycotoxigenic agents, could be applied for the degradation of mycotoxin occurrence hazards. Again, the cytotoxic impact of the polar extract was determined alone and with aflatoxin B1 using two types of cell-line. The promising extract was recommended for mycotoxin reduction in food products during food processing steps.

## 2. Results

### 2.1. Chemical Composition of Bottle Gourd Powder of the Seeds

The chemical composition analysis for the dried powder of the BG-seeds was determined in whole and defatted seeds to explore their contents, particularly of carbohydrates (active carbohydrates) and fiber contents (Table 1). The data represented valuable contents of both (carbohydrates and fibers) as well as minerals in the analyzed powder, concerning the defatted ones. Again, the results referred to the presence of oil residues in the dried powder from milled seeds. These residues may have participated in the impact of the extract for minimizing mycotoxin reduction. The defatted powder of seeds showed a high protein content, this protein has peptides fragments (Di, or polypeptides), which could interact with the fungal-metabolic cycle and leads to changes in the mycotoxin-secretion process. It is well known that fiber content is included oligo and polysaccharides; where members of these groups possess the ability to occur antifungal activity.

### 2.2. Fatty Acid Composition, Tocols, Sterols, and Carotenoids Content of Extracted Oil

The oil content of chemical compounds, which suggested having a bioactive function, was analyzed as four groups of oil content (Table 2). These groups were classified as fatty acid composition, vitamin E (tocols), sterols, and carotenoids. Each group of these chemical compounds could interact and influence the biological system of toxigenic fungi during their growth rate. For the fatty acid composition, oleic acid followed by linoleic acid act as the major fatty acid in the BG-oil.

Regarding the represented results in Table 2; the oil content of saturated fatty acid appears by a low ratio of percentage, the content of mono-unsaturated fatty acid recorded as the dominant amount, followed by the poly-unsaturated fatty acids. The ratio between the mono-unsaturated and the poly-unsaturated reported so-close to each other, and it was 5:4 between the two types of fatty acids groups.

### 2.3. Antioxidant Activity of the BG-Polar Extract

From this point forward, the authors suggested evaluating the polar extract as an anti-toxigenic agent with characteristics to reduce the mycotoxin secretion by fungal pathogens strains. The first parameter to evaluate was the antioxidant potency of the extract in comparison to a standard antioxidant (ascorbic acid). The results reflected significant values for the antioxidant of the polar extract using three assays of the evaluation (Figure 1).

### 2.4. Determination of Phenolic Fractions

The polar extract of bottle gourd seeds was shown as enrichment extract with phenolic fraction, these fractions were varied between phenolic acids and flavonoids (Table 3). The extract recorded a valuable content of phenolic acid, particularly ferulic acid followed by Chlorogenic acid and sinapic acid. While flavonoid compounds were presented by a majority for Apigenin followed by the catechin. However, the polar extract was recorded with a lower content of vanillic acid and protocatechuic acid. For the flavonoids, epicatechin and rutin were recorded as not detected in this type of extract. It is worth mentioning that; the content of ferulic as phenolic acid and apigenin were recorded so-close in the quantities. This point is taken into the consideration for their functional impact on the biological systems.

It is important to point out the valuable content of Chlorogenic acid, Sinapic acid, and 4-hydroxybenzoic acid as the dominant fractions represent the phenolic acid content of the polar extract. These phenolic acids may have a synergistic when these acids are present in the biological systems. Notwithstanding, flavonoids are represented by Apigenin and Catechin that be found in high amounts content, eight flavonoid components were alsofound despite their amounts being less than one microgram per gram of seed powder.

### 2.5. Antifungal Activities for the Polar Extract of Bottle Gourd

The antifungal activity of polar extract from the powder of bottle gourd seeds was evaluated using solid and liquid media. The solid media were applied to estimate the antifungal potency throughout the diffusion mechanism using well and disk diffusion assays. While liquid media was applied to estimate the extract influence to reduce the mycelial growth of fungi. The results represented in Figure 2, elucidated the inhibition impact of the extract for the mycelia and diffusion assays. The impact was major for *A. ochraceus* ITEM 2654 using liquid media for mycelia growth, while diffusion assays represent Fusarium fungi as the most influenced strain for the inhibition by the two assays (disk and well diffusion).

Throughout their life cycle, fungal cells engage in many transactions. The fungal suffering from stressors such as oxidative tension is linked to pivotal phases. This stress may cause fungal mutations, resulting in alterations in the cell life-hormone cycles and enzyme production. The major reason for toxin formation is oxidative stress, which is a crucial role in mycotoxin synthesis. The catalyst for mycotoxins production is oxidative stress in fungal cells.

### 2.6. Evaluation of Bottle Gourd Extract Healthy Impact

The impact of the polar extract was evaluated for two types of cell lines for evaluation of its effect on the viability of the cells (individually, or in the presence of aflatoxin). The result showed in Figure 3 reflects the safety impact of bottle gourd extract with low toxicity, and the efficiency of the extract to reduce the lethal dose of aflatoxin B1 when the polar extract was presented in the growth media of the cell line.

### 2.7. Reduction Impact of BG-Polar Extract for Mycotoxin Production

The impact of polar extract of bottle gourd seeds was shown by efficacy impact to reduce mycotoxin secretion when it was presented in the growth media (Figure 4). The efficacy impact was determined for the toxin secretion of *A. parasiticus* ITEM 10 that was known to produce aflatoxins. The results were reflected by a high reduction of zearalenone toxin followed by the reduction that occurred for aflatoxins. This result might be linked to the inhibition that happened to the fungal growth in liquid media.

The data represented in the above-mentioned results created a correlation between the bioactive components of the polar extract obtained from bottle gourd seeds and the mycotoxin reduction that happened when it was presented in the fungal growth media. The enrichment of the extract by antioxidants, phenolic fractions, and soluble contents of bioactive carbohydrates chemically supports its function to alter the fungal behavior throughout the fungal growth rates.

## 3. Discussion

Phytochemicals, particularly the phenolic fractions known to possess a significant influence on mycotoxin secretion by their presence within the fungi in the same environment [27]. The soluble content of oligosaccharides and dietary fibers is known to have an impact against toxigenic fungi and mycotoxins [28,29]. While mycotoxins can occur oxidative stress issues [30,31], the soluble content of seed extract that contains oligosaccharides and dietary fibers may serve against oxidative stress [32,33]. The oil was distinguished with unsaturated fatty acids, which were also known with degradation impact for mycotoxin secretion in some fungal bio-system [34]. However, these fatty acids were reported as a sporogenic factor for Aspergillus fungi [35], increasing spore formation (vegetative growth) that leads to less toxin production during fungi live-cycle. The second group was tocols (tocopherol and tocotrienol), which was recorded by significant amounts, particularly for gamma and delta fractions. These fractions are known to possess a biological function in the live systems [29,33,36].

Moreover, other substances of chemical components were presented in bottle gourd oil; it also was reported to have an impact against toxigenic fungal growth. These chemical substances are classified as sterols group and carotenoids group (Table 2). The majority was recorded for β-sitosterol of the sterols and the β-carotene of the carotenoids. Generally, the significant values of this oil reflect its nutritional characteristics according to the recommendation of the world health organization between the content of saturated and unsaturated fatty acids [20,22]. Otherwise, the antioxidant activity of the extract will play a vital role for decrease the environmental conditions influence on the fungi strain during the incubation, which consequently decreases the oxidative stress on the cell and leads to less mycotoxin production [21,36].

Phenolic fractions are the other components of bioactive substances, which have chemical and biological impacts against the secretion of mycotoxin by fungi during their life cycle [21,27]. The most abundant compound of the phenolic fractions was ferulic acid, which is known to degrade mycotoxin content [37,38,39,40]. It was reported previously that the extracts of the BG-seeds, which were obtained by ethanol and chloroform have antibacterial impact against four strains of pathogenic bacteria [41]. That study also reported the antioxidant activity of the BG-extracts and it was represented by high activity [41]. Other studies were referred to the phytochemical components of plant extracts as a valuable source, and it was possessed a function to achieve antimicrobial activity where it was applied [41,42]. The content of phenolic acids and flavonoids of the BG-extract using isopropanol showed distinguished amounts, this may be related to the extraction type [41,43]. Phenolic content is a bioactive constituent that connected to the antioxidant activity [42,43,44]. The types and the amount of these components reflect the degree of activity as antioxidant.

While the extracts with high antioxidant potency were reported to degrade mycotoxin amount in the growth media of fungi [16,36,45,46]. Moreover, the presence of bioactive carbohydrates represented by oligosaccharides and dietary fibers in the applied extract to the fungal growth media is considered a positive factor that leads to toxin amount decreased in the growth media of toxigenic fungi [28,32,33] throughout the reduction of the oxidative stress on the fungal cells. Again, the traces of oleic acid in the extract of bottle gourd seed may affect the behavior of fungal strain through the effect on sporulation (vegetative growth), which redirect the cell away from toxin production [35]. The previous research was pointed out the functionality of phenolic and flavonoids compounds as toxigenic fungi inhibitors reduce their growth in liquid media [15,17,45]. It was recorded by high activity by application of plant extraction in liquid media [27]. In this regard, more investigations are required to explain the mechanism of phenolic fractions influences when it was present in fungal growth media. Where aflatoxin is considered a carcinogenic chemical compound produced by toxigenic fungi, it was reported to decrease the viability of healthy cell-line [47,48].

The extract of the BG-extracts were reported as rich in their phytochemical content [41], which is in agreement with the present results in this investigation. Several prior studies have looked at the effect of plant phytochemicals on the aflatoxin gene cluster of toxigenic fungus and the aflatoxin biosynthesis pathway. In the presence of essential oils, for example, the expression of aflatoxin genes was observed to be downregulated. Furthermore, phytochemicals such as phenolic acids have been shown to inhibit gene expression in the aflatoxin pathway. Furthermore, phytochemicals from plants are important regulators of gene expression in toxigenic fungus.

While the fungal growth is occurred more by the normal environmental condition, and the toxin secretion is known to occur when these conditions are not suitable due to the oxidative stress on fungal cells [21,27]. The presence of bottle gourd extract could play a vital role to decrease the oxidative stress in the fungal media, leading to less mycotoxin secretion. Oxidative changes arrange several physiological processes in fungus cells, including the production of secondary metabolic components. The presence of oxidants and free radicals in the growth medium, in particular, is capable of promoting the biosynthesis of AF molecules. The interaction of the previous factors of the extract by its application in toxigenic fungal media is leading to the inhibition recorded by these investigations. Moreover, it was the principal factor for the mycotoxin secretion reduction in the fungal growth media.

## 4. Conclusions

Mycotoxin and their fungal-producing strains were the most harmful issue connected to the food product contamination. It was linked to the agrofood production stages both at pre and post-harvest time. It is significant to search for a valuable material that influenced the contamination reduction. Bottle gourd seeds are unique for their characteristics, with distinguished chemical composition. The seeds have two types of extracts (polar and non-polar), where the last one is recorded by significant contents of tocols, sterols, and carotenoids. The oleic acid and linoleic acid were reported by the majority content in this oil type. The polar extract reported higher content of Ferulic acid, this phenolic compound is known to have the efficiency to reduce mycotoxin production. The antioxidant potency of the extract could act function for the reduction impact of the extract. The application of the extract to reduce the lethal dose of aflatoxin is recorded by high efficiency against two types of cell lines. The impact of polar extract implementation in the fungal growth media of toxigenic strains reflects its potency for reducing the mycotoxin secretion by fungal cells. The authors recommended the application of polar extract as a factor that reduced mycotoxin contamination in raw food materials. This investigation was a part of a complete study for the bottle gourd extracts’ impact on toxigenic fungi and their chemical metabolites.

## 5. Materials and Methods

### 5.1. Chemicals, Reagents, and Sample Collection

Bottle gourd seeds (BGS) were collected from wild areas, the samples were physically cleaned to eliminate any extraneous particles like dust, stones, dirt, and immature seeds before being mixed to give about 2 kg of seeds. Before utilization, the seeds were kept in an airtight plastic container and maintained at 4 °C. The seeds were allowed to acclimatize to equilibrium moisture under ambient room settings (20–22 °C, 30%–35% RH) before commencing the experiment.

Chemicals and reagents that were utilized in this experiment were purchased from Fluka(™) chemicals analytical brand, distributed by fisher scientific, ENA 23, Zone 1, nr 1350, Janssen Pharmaceuticalaan 3a, 2440 Geel, Belgium. The standards that were used for the comparative evaluations were purchased from Sigma-Aldrich(™) chemicals, Sigma-Aldrich Chemie GmbH, Eschenstr. 5, 82024 Taufkirchen, Germany.

### 5.2. Preparation of Polar and Non-Polar Extracts

Two types of extracts were prepared for bottle gourd seeds to determine the most effective extract, which possesses an anti-mycotic and anti-mycotoxigenic impact. The seeds were milled using Laboratory Mill 3310 (Seedburo, Chicago, IL, USA) to get fine powder used for extract preparation. The polar extract was prepared as described by Abdel-Razek et al., [46] with fine modification. A constant weight of seeds powder was mixed by aqueous isopropyl solution (5:1 *w*/*v*) in a Conical flask and treated by ultrasonication (Power amplitude 45%, frequency 80 kHz, duty cycle 60%, time cycle 45 min, and 22 °C).

### 5.3. Determination of Total Phenolic and Flavonoid Contents (TPC)

The total phenolic content (TPC) of a polar extract was determined according to the methodology described by Badr et al., [19], while the non-polar extract was determined according to the methodology described by Badr et al. [21]. The absorbance was measured using a UV-spectrophotometer (Shimadzu, Kyoto, Japan) at wavelength 760 nm. The TPC was measured and expressed as mg gallic acid equivalent (GAE)/100 g of the sample against the blank.

The total flavonoid content (TFC) of the PGO was determined using an aluminum chloride colorimetric method [49]. Briefly, 1 mL aqueous methanol extract (1 mg/mL; 80% methanol) or standard catechol solution was mixed with one milliliter of the AlCl3 (2% *w*/*v*) in methanol. The absorbance against blank was measured at 440 nm using the UV spectrophotometer (after incubating for 40 min/25 °C). The TFCs were calculated using the catechol calibration curve; results were expressed as catechol equivalent (CE)/g of sample, where the samples were performed in triplicate.

### 5.4. Determination of Fatty Acid Profile

The fatty acid content was evaluated using the methodology described by Abdel-Razek et al. [20]. In brief, the Agilent 7890 apparatus (Agilent Technologies, Santa Clara, CA, USA) was used to evaluate diluted oil, which was supported by the FID and capillary Innowax column (30 m × 0.20 mm × 0.20 mm). The carrier gas flow rate was 1.5 mL/min, and the temperature of the column was 210 °C. After integrating and calculating the findings using the Chem-Station and comparing the retention durations to genuine standards, the results were reported as weight percentages.

### 5.5. Determination of Tocopherol and Tocotrienol Contents

The technique established by Balz et al. [50] was used to assess tocopherol and tocotrienol. The mobile phase was n-hexane:tertabutyl-methyl ether (96:4 *v/v*), the isocratic system was used with 1.0 mL/min flow and a wavelength of 295 nm, and the UV detector was used. The results were determined in triplicates and expressed as means ± SEM.

### 5.6. Determination of Sterols and Carotenoid Content

The sterols content was determined as the methodology substantive in Stuper-Szablewska et al. [51]. The carotenoids were determined utilizing the Acquity ultra-high performance liquid chromatography (Waters, Milford, MA, USA) as the same methodology and conditions substantive in Kurasiak-Popowska et al. [52].

### 5.7. Antioxidant Activity for Polar Extract

#### 5.7.1. Scavenging Activity of DPPH Radicals

Antioxidant activity was determined using the 2,2-diphenyl-1-picrylhydrazyl (DPPH) free radical scavenging assay according to the methodology described by Abdel-Razek et al., [46] in the polar extract. The absorbance was measured at 517 nm using a UV spectrophotometer. The tests were carried out in triplicates. The DPPH radical inhibition is calculated using the following equation:%Inhibition = [Abs_517_ (control) − Abs _517_ (sample)/Abs _517_ (control)] × 100 
where;Abs _517_ (control): The absorbance of the blank at 517 wavelengths.Abs _517_ (sample): The absorbance of the determined sample at 517 wavelengths.

#### 5.7.2. Determination of Reducing Power

The reducing power of extracts was determined according to the method of Yen and Duh [53], which was modified by Badr et al., [21]. The absorbance was measured at 700 nm, and reducing power was expressed as ASE/mg. The ASE means equivalent power of 1 mg sample (E) reducing the power of 1 nmol ascorbic acid (AS).

#### 5.7.3. Antioxidant Activities Using Scavenging of Hydroxyl Radicals

Hydroxyl radical scavenging (HRS) was calculated using the Fenton reaction assay, which was elucidated by Shehata et al. [54]. Briefly, a reaction mixture containing Brilliant Green (0.435 mM), FeSO_4_, (0.5 mM), H_2_O_2_, (3.0%, *w*/*v*), and bacterial lysate extract in different concentrations was incubated at room temperature for 20 min, where the absorbance was measured at 625 nm. The changes in absorbance were referred to scavenging ability of bacterial strains for hydroxyl radicals. The HRS activity was expressed as follows:Scavenging activity (%) = (A_S_ − A_0_)/(A − A_0_) × 100 
where:A_S_: Is the absorbance of the sampleA_0_: Is the absorbance of the controlA: Is the absorbance without sample or Fenton reaction system.

### 5.8. Determination of Phenolic Fractions of Polar Extract

The phenolic fraction compounds in the Bottle gourd aqueous methanolic extract were determined according to the methodology described by Stuper-Szablewska et al. [51]. The phenolic compound concentrations were determined at λ = 320 and 280 nm and were identified based on a retention time comparison of analyte peaks by a retention time for standards and by adding a specific amount of the standard to the analyzed samples and repeating this analysis. The limit of detection was 1 µg/g sample, where the evaluation was done in triplicates and expressed as means ± SEM.

### 5.9. Estimation of Antifungal Activity of the Polar BG Extract

#### 5.9.1. Agar Diffusion Assay

Antifungal activity of polar BG extracts was screened by agar diffusion assay (disc and wells) as described in CLSI methodology (CLSI, 2012). Applied strains of fungi were *Aspergillus parasiticus* ITEM 10, *Aspergillus ochraceus* ITEM 2456, *Penicillium verricosum*, and *Fusarium culmorum* KF191. These strains were selected as strain-producing fungi used for secretion-reduction evaluation of mycotoxin; and were obtained from the Agro-Food Microbial Culture Collection of the Institute of Sciences and Food Production (ISPA, Bari, Italy). Fungal strains were reactivated on Czapek-Dox agar supplemented with tetracycline to suppress bacterial growth contamination. The effect is expressed as an inhibition zone (mm) surrounding the disk or the well.

#### 5.9.2. Fungal-Growth Inhibition

The liquid media was utilized for evaluating the reduction of fungal growth due to the presence of BG-polar extract following the methodology of Shehata et al., [54]. In brief, media of Czapek-Dox for Aspergillus parasiticus ITEM 10, Aspergillus ochraceus ITEM 2456, Penicillium verricosum, and Fusarium culmorum KF191. The 1 L-capacity flasks were contained 250 mL media for applying each treatment. The fungi-treated flasks were divided into six groups, where the first serving as the control group contained fungi alone (control negative) and the second contained fungi and Amphotericin B (30 µg/mL; control positive). The remains were classified as a group for each fungal strain. The inhibition growth of each fungus was calculated as follow:$$\mathrm{Fungal}\;\mathrm{inhibition}\;\left(\%\right)=\frac{\mathrm{control}\;\mathrm{growth}-\mathrm{treated}\;\mathrm{growth}}{\mathrm{control}\;\mathrm{growth}}\times 100$$
where,Control: media just contained inoculated fungiTreated: media contains the BG extract with inoculated fungi.

The extract was applied at concentrations of 300 µg/mL media because of lab-experimental investigations of the minimal inhibition concentration was varied between 250–280 µg/mL against tested strains of toxigenic fungi.

#### 5.9.3. Toxin Production Reduction in Liquid Media

In the previous amount of liquid media, and after the filtration of mycelial growth for the growth inhibition evaluation, the liquid filtrates (of control and BG-treatment) were utilized for mycotoxin reduction-evaluation as the methodology described before [54]. The toxin amount in liquid media was determined in triplicates, where the data were represented as means ± SEM.

### 5.10. Evaluation of Bottle Gourd Extract Healthy Impact

The presence of AFB1 reduced cell viability in a time and concentration-dependent pattern, according to the findings [48]. The authors proposed that healthy cells be exposed to AFB1 concentrations (with or without BG—extract). Human hepatocellular carcinoma (HepG2) and human epithelial colorectal adenocarcinoma (Caco-2) cells were grown in cell culture flasks (vented cap 75 cm^2^ ClearLine^®^ sterile) at 37 °C and 95% relative humidity in a 5% carbon dioxide environment. HepG2 and Caco-2 cell lines were maintained in MEM medium supplemented with 10% foetal bovine serum, 1% penicillin-streptomycin, 1 mM sodium pyruvate, and 2 mM L-glutamine. The adherent cells were dissociated from the flask with TrypLETM Express before being stained with trypan blue for cell counting and viability tests using a Countess^®^ automatic cell counter.

The same conditions and concentrations were used to seed the cells as before [55]. Cells were treated to five doses of single and mix of the test single or mixed chemicals (AFB1, BGE, AFB1-BGE) for 48 h after being exposed to them for 24 h. Negative controls included 0.5 percent (*v/v*) methanol/media and 0.1 percent (*v/v*) DMSO/media, while positive controls included valinomycin (in DMSO) at 60 M (0.1 percent (*v/v*) DMSO/media). The values of IC50 for the applied materials also were calculated. All samples were examined in triplicates.

### 5.11. Aflatoxins and Zearalenone Determinations

High-performance liquid chromatography, Agilent 1100 (Agilent Technologies, Hewlett-Packard Strasse 876,337 Waldbronn, Germany), was used for AFs determination. The mobile phase was water: acetonitrile: methanol (6:3:1). The chromatographic separation was performed with an Extend-C18, Zorbax column (250 mm × 5 µm, Agilent Co., Carpinteria, CA, USA). The column temperature was 40 °C and the flow rate was 1.0 mL/min; the injection volume was 20 L for samples and standard. The detector adjusted at 360/440 nm for the excitation and the emission wavelength, respectively. Data were integrated and recorded using a Chem-Station software Manager Hewlett-Packard.

Regarding zearalenone, it was isolated and analyzed from media using procedures, apparatus, and conditions described in Badr et al. [21,56]. The mobile phase was a 46:46:8 *v/v* combination of acetonitrile, water, and methanol, with a flow rate of 1.0 mL/min. The retention period of the ZEA was compared to a standard to determine its quantity. When compared to the ZEA standard peak, the identification of the ZEA was verified at excitation and emission wavelengths of 274 and 440 nm, respectively.

### 5.12. Statistical Analysis

The results were expressed as mean values standard deviation from at least three replicates. The statistical analyses of data were performed using Graph Pad Prism 7 (Graph Pad Software Inc., San Diego, CA, USA).

## Figures and Tables

**Figure 1 toxins-13-00789-f001:**
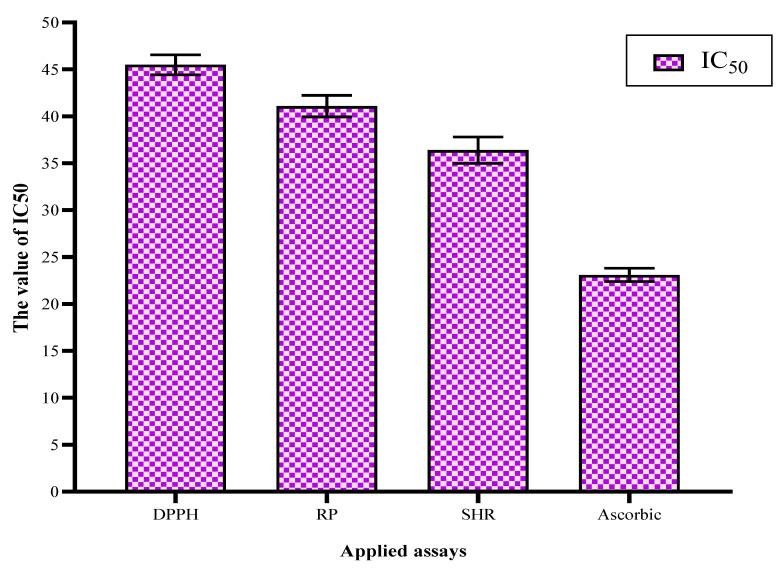
Antioxidant activities of bottle gourd polar extract using several assays.

**Figure 2 toxins-13-00789-f002:**
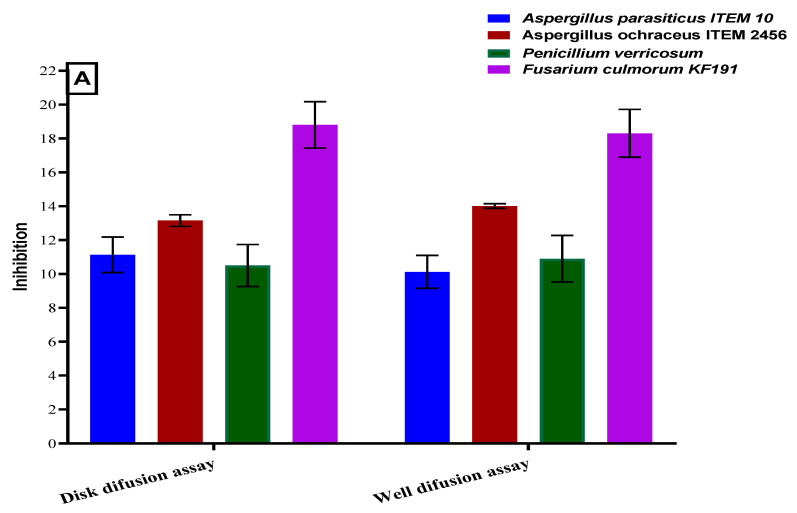
Antifungal potency of seed polar extract of bottle gourd. (**A**): antifungal potency evaluated using agar diffusion assays. (**B**): antifungal potency evaluated as mycelial inhibition using liquid media.

**Figure 3 toxins-13-00789-f003:**
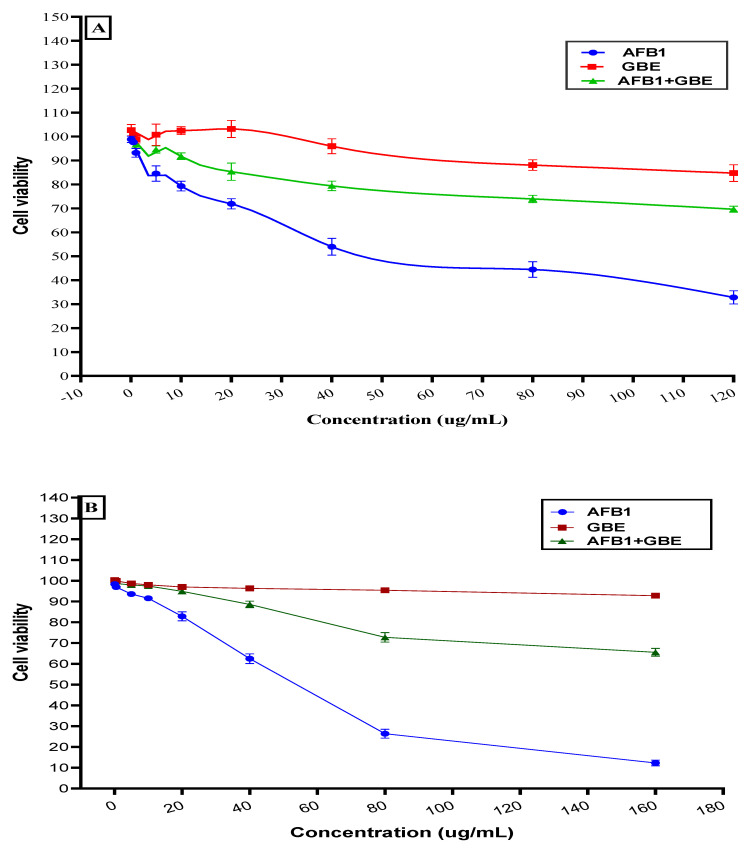
The seed polar extract of bottle gourd impact on cell-line viability (with/without aflatoxin), (**A**): the impact on the HEP-G cell line viability. (**B**): the impact on Caco2 cell line viability.

**Figure 4 toxins-13-00789-f004:**
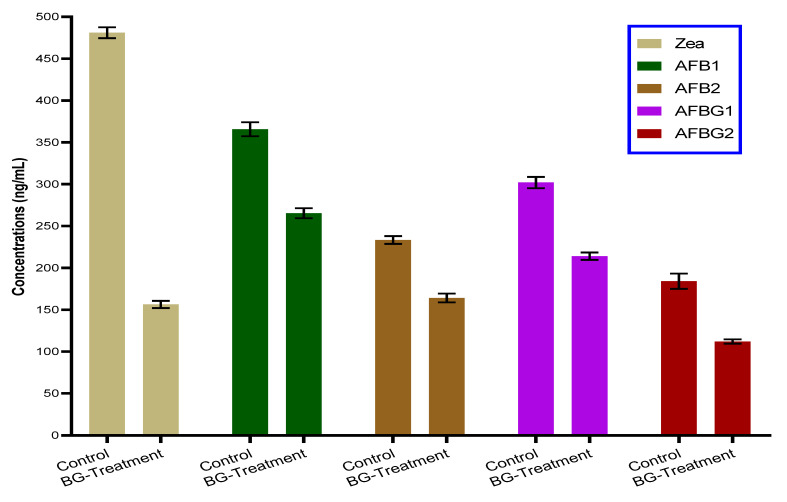
The seed polar extract of bottle gourd impact on mycotoxin secretion in liquid media of fungal growth.

**Table 1 toxins-13-00789-t001:** Chemical composition of whole and defatted powder of bottle gourd seeds.

Material	Moisture	Fat	Total Protein	Total Carbohydrates	Ash	Crude Fiber
Seed powder	4.92 ± 0.93 ^a^	34.12 ± 1.02 ^a^	11.54 ± 1.05 ^a^	19.21 ± 1.14 ^a^	2.51 ± 0.73 ^a^	32.41 ± 1.54 ^a^
de-fatted powder	6.37 ± 0.57 ^b^	2.17 ± 0.34 ^b^	18.61 ± 1.27 ^b^	31.18 ± 1.31 ^b^	4.27 ± 0.88 ^b^	52.11 ± 1.74 ^b^

The results are represented as means ± SEM, where (*n* = 3). The values represented by different superscripted letters for the same column are significantly different.

**Table 2 toxins-13-00789-t002:** Chemical constituents of oil extracted from the bottle gourd seeds.

Fatty Acid	Concentration (%)	Tocols Compounds	µg/g
C16:0 (palmitic acid)	6.41 ± 0.41	α-tocopherol	367.18 ± 4.26
C16:1 (palmitoleic acid)	0.39 ± 0.61	β-tocopherol	ND
C17:0 (heptadecanoic acid)	ND	γ-tocopherol	64.09 ± 2.14
C17:1 (heptadecenoic acid)	ND	δ-tocopherol	117.85 ± 3.71
C18:0 (stearic acid)	0.38 ± 0.18	Total tocopherol	549.12 ± 10.11
C18:1 (oleic acid)	49.37 ± 1.14		**µg/g**
C18:2n6 (linoleic acid)	35.21 ± 1.05	α-tocotrienol	27.37 ± 1.67
C18:3n6 (y-linolenic acid)	7.51 ± 0.54	β-tocotrienol	1.25 ± 0.34
C18:3n3 (linolenic acid)	0.22 ± 0.08	γ-tocotrienol	45.61 ± 1.46
C20:0 (arachidic acid)	0.10 ± 0.02	δ-tocotrienol	18.12 ± 1.02
C20:1 (c-11-eicosenoic acid)	ND	Total tocotrienol	92.35 ± 4.49
C20:2 (eicosadienoic acid)	ND		
C21:0 (heneicosanoic acid)	ND	**Sterols**	**mg/100 g**
C22:1 (erukowy)	0.18 ± 0.01	Campesterol	59.61 ± 2.05
C24:0 (tetrakozanowy)	0.20 ± 0.02	Stigmasterol	6.74 ± 0.54
C24:1 (nerwonowy)	ND	Ergosterol	22.40 ± 1.46
		β-sitosterol	332.66 ± 5.71
**Oil-Significant values**		δ-5-avenasterol	1.94 ± 0.22
SFA	7.09		
MUFA	50.51	**Carotenoids**	**µg/g**
PUFA	42.94	lutein	109.78 ± 2.66
SFA:MUFA:PUFA	0.21:1.51:1.28	Zeaxanthin	294.24 ± 3.08
Cox value of seeds oil	5.8	β-carotene	674.16 ± 5.74

The results are represented as means ± SEM, where (*n* = 3). Cox value = [1 * (18:1%) + 10.3 * (18:2%) + 21.6 * (18:3%)]/100.

**Table 3 toxins-13-00789-t003:** Chemical constituents of phenolic fractions of the polar extract from bottle gourd seeds powder.

Phenolic Acids	Concentrations in Polar Extract (µg/g)	Flavonoids	Concentrations in Polar Extract (µg/g)
Chlorogenic	97.15 ± 1.58	Apigenin	105.3 ± 2.54
Syringic	6.25 ± 0.51	Catechin	45.2 ± 1.46
4-hydroxybenzoic	71.6 ± 1.05	Epicatechin	ND
Caffeic	5.22 ± 0.41	Luteolin	0.19 ± 0.05
Ferulic	105.2 ± 2.88	Rutin	ND
Gallic	14.3 ± 0.97	Naringin	0.28 ± 0.06
*p*-Cumaric	52.1 ± 1.81	Quercetin	ND
Protocatechuic	0.3 ± 0.02	Apigenin-7-glucoside	0.31 ± 0.14
Sinapic	91.2 ± 1.37	Kaempferol	6.25 ± 0.51
Vanilic	0.56 ± 0.22	Chrysin	0.21 ± 0.03

The results are represented as means ± SEM, where (*n* = 3).

## Data Availability

The data in this study are available in this article.

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
