# Peer review of "Efficacy of Bottle Gourd Seeds’ Extracts in Chemical Hazard Reduction Secreted as Toxigenic Fungi Metabolites"

_toxins, 2021, doi:10.3390/toxins13110789_

Round 1

Reviewer 1 Report

The manuscript "Assessment of Bioactive Bottle Gourd Seeds' Constituents to reduce the chemical hazard caused by toxigenic fungi metabolites" is an interesting study carried out at a good methodological level. The results are of fundamental importance, and the authors also provide practical recommendations.

In my opinion, the title of the article is not very successful and does not reflect the whole essence of the work performed.
The introduction is written in detail, but contains a fragment of instructions for writing it.
The results are described well, but the quality of the pictures, especially fig. 2, is very poor
The name of the fungi should be italicized.
It would also be good to explain the choice of these particular species as test objects and give a more detailed characterization of the strains.
Section 2.7 should mention the fungal species in question.
The article is interesting and contains important results, but it gives a sloppy impression and needs further revision.

Author Response

Reviewer 1:

The manuscript "Assessment of Bioactive Bottle Gourd Seeds' Constituents to reduce the chemical hazard caused by toxigenic fungi metabolites" is an interesting study carried out at a good methodological level. The results are of fundamental importance, and the authors also provide practical recommendations.

In my opinion, the title of the article is not very successful and does not reflect the whole essence of the work performed.

Response: this point was done, and the title was changed to be

Efficacy of Bottle Gourd Seeds’ extracts in chemical hazard reduction secreted as toxigenic fungi metabolites.

The introduction is written in detail but contains a fragment of instructions for writing it.

Response: this point was done.

The results are described well, but the quality of the pictures, especially fig. 2, is very poor
The name of the fungi should be italicized.

Response: this point was corrected as requested.

It would also be good to explain the choice of these particular species as test objects and give a more detailed characterization of the strains.

Response: this point was done in the section of materials and methods.

Section 2.7 should mention the fungal species in question.

Response: this point was done in the referred section.

The article is interesting and contains important results, but it gives a sloppy impression and needs further revision.

Response:  Thanks for your efforts in the manuscript revision; this point was done.

Reviewer 2 Report

Dear authors. The work sent to me for review is very interesting and interesting. However, it requires the necessary correction. Here are some tips for improving your work. 

In the introduction, lines 30-38 should be marked with information about the journal's requirements, it is unnecessary. 

The sentence "According to the WHO, traditional medicines are the best and most cost-effective source of anti-infection and anti-cancer substances [5] lines 48-49 - should be removed because it does not relate to the topic of the research conducted. 

Line 71-78 Bottle gourd (BG) is one of the veggies and calorie-dense that dieticians recommend in weight-loss regimens [23]. The spongy flesh tissues along with white pulp and embedded seeds exist inside the bottle gourd fruits. The seeds are present in large numbers and all are covered with a protectant layer. Seeds are distinguished by their content of protein, fats, dietary fibers, and low carbohydrates [24, 25]. Seed kernels mostly yield up to 53 % oil as clear and pale yellowish used for cooking and hair oil. The flour of seeds is considered a significant source for vitamins (B-complex vitamins) and minerals inclusive of potassium, calcium, zinc, magnesium, iron, and manganese [25]. Discard because the human repair value is not relevant for mycotoxin protection. Instead, more detailed information should be provided on the importance of the compounds used in the research and their use in protecting against the production of mycotoxins by fungi. 

Lines 87-93, these results and tables should be removed. It does not contribute any relevant information to the conducted research and is not related to its purpose.

In the following, the results are described very briefly. Expand the description of the tables and the results obtained. In such a short form, they add nothing to the results. 

Discussions should be expanded. It is too short and does not relate to the results of the research obtained in this work and does not compare the results obtained by other researchers. I propose to use the following literature: Amin, T., Naik, H. R., Hussain, S. Z., Jabeen, A., & Thakur, M. (2018). In-vitro antioxidant and antibacterial activities of pumpkin, quince, muskmelon and bottle gourd seeds. Journal of Food Measurement and Characterization, 12 (1), 182-190. Antia, B. S., Essien, E. E., & Udoh, B. I. (2015). Antioxidant capacity of phenolic from seed extracts of Lagenaria siceraria (short-hybrid bottle gourd). European Journal of Medicinal Plants, 1-9. Bhat, S., Saini, C. S., & Sharma, H. K. (2017). Changes in total phenolic content and color of bottle gourd (Lagenaria siceraria) juice upon conventional and ohmic blanching. Food science and biotechnology, 26 (1), 29-36. Attar, U. A., & Ghane, S. G. (2019). In vitro antioxidant, antidiabetic, antiacetylcholine esterase, anticancer activities and RP-HPLC analysis of phenolics from the wild bottle gourd (Lagenaria siceraria (Molina) Standl.). South African Journal of Botany, 125, 360-370. Essien, E. E., Antia, B. S., & Peter, N. S. (2015). Lagenaria siceraria (molin) standley. total polyphenols and antioxidant activity of seed oils of bottle gourd cultivars. World J. Pharmaceut. Res. 4 (6), 274-285. Minocha, S. (2015). An overview on Lagenaria siceraria (bottle gourd). Journal of Biomedical and Pharmaceutical Research, 4 (3), 4-10. 

Author Response

Reviewer 2:

Dear authors. The work sent to me for review is very interesting and interesting. However, it requires the necessary correction. Here are some tips for improving your work. 

In the introduction, lines 30-38 should be marked with information about the journal's requirements, it is unnecessary. 

Response: this point was deleted thanks, it was Typographic error.

The sentence "According to the WHO, traditional medicines are the best and most cost-effective source of anti-infection and anti-cancer substances [5] lines 48-49 - should be removed because it does not relate to the topic of the research conducted. 

Response: this point was re-written thanks

Line 71-78 Bottle gourd (BG) is one of the veggies and calorie-dense that dieticians recommend in weight-loss regimens [23]. The spongy flesh tissues along with white pulp and embedded seeds exist inside the bottle gourd fruits. The seeds are present in large numbers and all are covered with a protectant layer. Seeds are distinguished by their content of protein, fats, dietary fibers, and low carbohydrates [24, 25]. Seed kernels mostly yield up to 53 % oil as clear and pale yellowish used for cooking and hair oil. The flour of seeds is considered a significant source for vitamins (B-complex vitamins) and minerals inclusive of potassium, calcium, zinc, magnesium, iron, and manganese [25]. Discard because the human repair value is not relevant for mycotoxin protection. Instead, more detailed information should be provided on the importance of the compounds used in the research and their use in protecting against the production of mycotoxins by fungi. 

Response: This paragraph was added here to describe the role of seed extracts disscused in the discussion section.

Lines 87-93, these results and tables should be removed. It does not contribute any relevant information to the conducted research and is not related to its purpose.

Response: This part was added here to describe the co-operative function of seed oil residues interacting in the secretion reduction; however, this study consist of two-part, and more details will be further discussed.

In the following, the results are described very briefly. Expand the description of the tables and the results obtained. In such a short form, they add nothing to the results.

Discussions should be expanded. It is too short and does not relate to the results of the research obtained in this work and does not compare the results obtained by other researchers. I propose to use the following literature: Amin, T., Naik, H. R., Hussain, S. Z., Jabeen, A., & Thakur, M. (2018). In-vitro antioxidant and antibacterial activities of pumpkin, quince, muskmelon and bottle gourd seeds. Journal of Food Measurement and Characterization, 12 (1), 182-190. Antia, B. S., Essien, E. E., & Udoh, B. I. (2015). Antioxidant capacity of phenolic from seed extracts of Lagenaria siceraria (short-hybrid bottle gourd). European Journal of Medicinal Plants, 1-9. Bhat, S., Saini, C. S., & Sharma, H. K. (2017). Changes in total phenolic content and color of bottle gourd (Lagenaria siceraria) juice upon conventional and ohmic blanching. Food science and biotechnology, 26 (1), 29-36. Attar, U. A., & Ghane, S. G. (2019). In vitro antioxidant, antidiabetic, antiacetylcholine esterase, anticancer activities and RP-HPLC analysis of phenolics from the wild bottle gourd (Lagenaria siceraria (Molina) Standl.). South African Journal of Botany, 125, 360-370. Essien, E. E., Antia, B. S., & Peter, N. S. (2015). Lagenaria siceraria (molin) standley. total polyphenols and antioxidant activity of seed oils of bottle gourd cultivars. World J. Pharmaceut. Res. 4 (6), 274-285. Minocha, S. (2015). An overview on Lagenaria siceraria (bottle gourd). Journal of Biomedical and Pharmaceutical Research, 4 (3), 4-10. 

Response: this point was done.

Round 2

Reviewer 2 Report

Dear Authors,

Thank you for the corrections and clarifications introduced. The work in its current form looks very good and meets the requirements of the scientific work.